# An Empirical Study on Multiple Knowledge from ChatGPT for Emotion Recognition in Conversations

**Geng Tu**[1,2], **Bin Liang**[3] *, **Bing Qin**[4], **Kam-Fai Wong**[3], **Ruifeng Xu**[1,2,5] *

[1]Harbin Institute of Technology, Shenzhen, China
[2]Guangdong Provincial Key Laboratory of Novel Security Intelligence Technologies
[3]The Chinese University of Hong Kong, Hong Kong, China
[4]Harbin Institute of Technology, Harbin, China
[5]Peng Cheng Laboratory, Shenzhen, China

tugeng0313@gmail.com, bin.liang@cuhk.edu.hk, xuruifeng@hit.edu.cn

## Abstract

Multiple knowledge (e.g., co-reference, topics, emotional causes, etc) has been demonstrated effective for emotion detection. However, exploring this knowledge in Emotion Recognition in Conversations (ERC) is currently a blank slate due to the lack of annotated data and the high cost involved in obtaining such knowledge. Fortunately, the emergence of Large Language Models (LLMs) holds promise in filling this void. Therefore, we propose a Multiple Knowledge Fusion Model (MKFM) to effectively integrate such knowledge generated by LLMs for ERC and empirically study its impact on the model. Experimental results on three public datasets have demonstrated the effectiveness of multiple knowledge for ERC. Furthermore, we conduct a detailed analysis of the contribution and complementarity of this knowledge[1].

## 1 Introduction

Emotion recognition in conversations (ERC) has garnered considerable research interest in recent years, as evidenced by several studies such as (Tu et al., 2022b; Xie et al., 2021; Mao et al., 2021; Tu et al., 2023b). The goal of ERC is to identify the emotion of each utterance in conversations.

Existing efforts in ERC have traditionally focused on modeling context- and speaker-sensitive dependencies (Lian et al., 2021), including recurrent-based network (Hazarika et al., 2018; Majumder et al., 2019; Ghosal et al., 2020; Jiao et al., 2020; Li et al., 2022b), transformer-based network (Lian et al., 2021; Shen et al., 2021a; Ong et al., 2022), and graph-based network (Ghosal et al., 2019; Shen et al., 2021b; Saxena et al., 2022). Additionally, recent ERC models (Tu et al., 2022a; Ghosal et al., 2020; Zhong et al., 2019; Jiang et al., 2022) have begun to leverage commonsense knowledge, which mainly includes two categories. One is

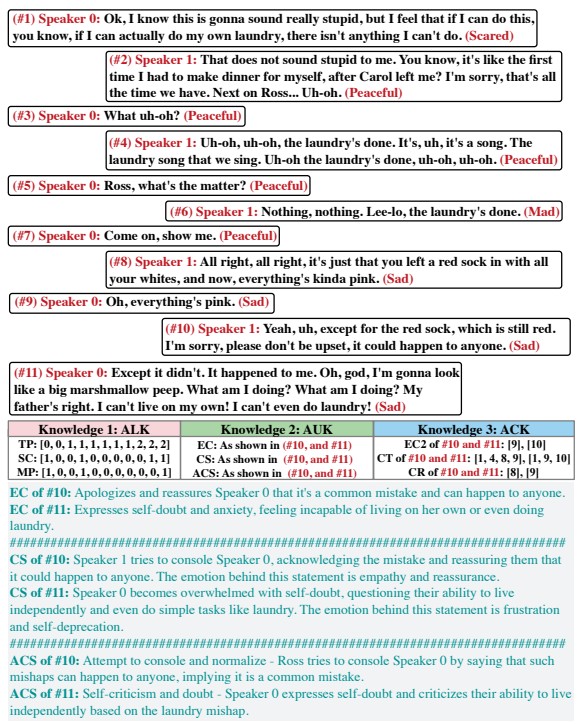

Figure 1: Examples of utterances, reflecting the process of introducing multiple knowledge for ERC.

generative knowledge generated by the pre-trained commonsense transformer (COMET) (Bosselut et al., 2019). The other is concepts retrieved from the external knowledge base ConceptNet (Speer et al., 2017) or SenticNet (Cambria et al., 2020).

However, relying solely on commonsense knowledge as background information (Bauer et al., 2018) is insufficient for a deep understanding and recognition of emotions. Multiple knowledge (e.g., co-reference, topics, emotional causes, etc) are also crucial for emotion detection, but integrating this knowledge into the ERC model poses challenges due to the limited availability of annotated data and the high cost of manual tagging. Fortunately, Large Language Models (LLMs) such as ChatGPT provides an opportunity to overcome these limitations, as illustrated in Fig. 1. We divide this knowledge

---

* Corresponding authors.

[1]The code is available at https://github.com/TuGengs/MKFM.

from ChatGPT into three categories based on data formats, namely Auxiliary Contextual Knowledge (ACK with index list format): **Co-reference (CR)** is crucial for modeling multi-party dialogue in improving the dialogue understanding (Li and Zhao, 2021); **Emotional Cause (EC2)** reasoning involves identifying the reasons behind emotions and how contextual utterances contribute to them (Poria et al., 2021); **Context (CT)** is indicated as important in emotion generation theory (Gross and Barrett, 2011), Auxiliary Label Knowledge (ALK with label format): **Topics (TP)** play a significant role as emotions are closely tied to specific subjects, and the same words can have different emotional meanings depending on the topic (Zhu et al., 2021); **Sarcasm (SC)** is an important linguistic tool that uses irony to express contempt and affects the accurate prediction of emotional meaning (Poria et al., 2019b); **Metaphor (MP)** is common language expression that conveys nonliteral meanings and emotions by comparing or connecting concepts (Mohammad et al., 2016), and Auxiliary Utterance Knowledge (AUK with sentence format): Commonsense knowledge (CS) related to the utterances; Affective commonsense knowledge (ACS) that bridges the cognitive and affective gap between word-level natural language data and the concept-level sentiments conveyed by them (Cambria et al., 2012); Another format of **Emotional Cause (EC)**. Then we introduce a Multiple Knowledge Fusion Model (MKFM) that combines three types of knowledge via utterance-level encoder for AUK, graph context encoder for ACK, and Supervised Contrastive Learning (SCL) module for ALK.

Our main contributions are (1) the first exploration of multiple knowledge for ERC, (2) the proposed MKFM for integrating this knowledge, which outperforms state-of-the-art baselines, and (3) conducting further analysis on the contribution and complementarity of multiple knowledge.

## 2  Related Work

**Emotion Recognition in Conversations:** The emotion generation theory (Gross and Barrett, 2011) emphasizes the role of contextual information in identifying emotions. RNN-based models (Poria et al., 2017) are commonly used to capture context dependencies but struggle with distinguishing between historical utterances (Lian et al., 2021). Memory networks have been proposed to address this limitation (Hazarika et al., 2018; Jiao

et al., 2020). Furthermore, the participants in ERC are crucial (Wen et al., 2023), leading to the development of speaker-specific models (Kim and Vossen, 2021; Majumder et al., 2019; Ghosal et al., 2020), graph-based models (Nie et al., 2021; Shen et al., 2021b; Ghosal et al., 2019), and so on. However, these approaches lack commonsense knowledge, which is important for human-like performance (Tu et al., 2023a). Therefore, researchers have integrated external knowledge sources like COMET (Bosselut et al., 2019), SenticNet (Cambria et al., 2022), and ConceptNet (Speer et al., 2017) into their models (Zhao et al., 2022; Ghosal et al., 2020; Fu et al., 2021; Jiang et al., 2022).

**Contrastive Learning:** Chen et al. (2020) introduced SimCLR, a comparative learning network for visual representation using image augmentation. In NLP, Yan et al. (2021) proposed a self-supervised CL method for fine-tuning BERT, addressing BERT's poor performance in semantic text similarity tasks. To incorporate label information, Gunel et al. (2020) extended self-supervised CL to a fully-supervised CL framework, which improved performance in few-shot learning scenarios. In ERC, Li et al. (2022a) and Song et al. (2022) applied SCL to separate utterances with different emotions, enhancing emotion identification. However, there is no effort on unsupervised CL in ERC.

## 3  Methodology

### 3.1  Task Definition

Let $\mathbf{C}$ be a conversation consisting of $n$ utterances. Each utterance $\mathbf{u}_i$ is spoken by one of the speakers in the set $\mathbf{S}$, which contains $m$ speakers: $\{\mathbf{s}_1, \mathbf{s}_2, ..., \mathbf{s}_m\}$. The objective of ERC is to predict the emotion label $\mathbf{y}_i$ for each utterance $\mathbf{u}_i$.

### 3.2  Overview

In the MKFM shown in Fig. 2, we incorporate multiple knowledge. Based on how the model integrates these knowledge types, we categorize them into three classes. In the following sections, we provide detailed descriptions of each module.

### 3.3  Utterance-level Encoder

Following Ghosal et al. (2020), we utilize the pretrained model RoBERTa (Liu et al., 2019) to encode the input vector $\mathbf{u}_i$. To enhance the utterance representations using AUK, we include this knowl-

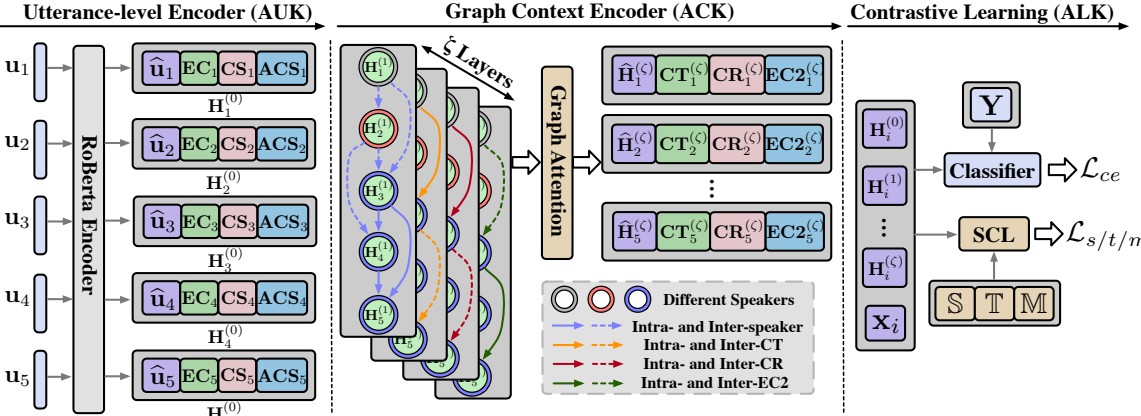

Figure 2: Illustration of MKFM, where mathematical symbols are consistent with the formulas in this paper.

edge by utilizing the concatenation operation.

$$\mathbf{H}_i^{(0)} = \mathbf{Linear}(\mathbf{x}_i) \quad (1)$$

$$\mathbf{x}_i = \widehat{\mathbf{u}}_i \oplus \mathbf{EC}_i \oplus \mathbf{CS}_i \oplus \mathbf{ACS}_i \quad (2)$$

$$\widehat{\mathbf{u}}_i = \mathbf{RoBERTa}(\mathbf{u}_i) \quad (3)$$

where $\widehat{\mathbf{u}}_i$ is the $d_u$ dimension hidden states of $\mathbf{u}_i$. $\mathbf{H}_i^{(0)} \in \mathbb{R}^{d_h}$ is the vector representation after linear transformation $\mathbf{Linear}$. $\oplus$ represents the concatenation operation. $\mathbf{EC}_i \in \mathbb{R}^{d_u}$ represents the emotion cause of the $i$-th utterance, $\mathbf{CS}_i \in \mathbb{R}^{d_u}$ represents the commonsense knowledge, and $\mathbf{ACS}_i \in \mathbb{R}^{d_u}$ represents the affective commonsense knowledge.

## 3.4 Graph Context Encoder

In ERC, extracting contextual information from surrounding utterances is crucial. However, relying on future utterances to determine the emotion of the current utterance is not practical in real-life situations (Poria et al., 2017; Ghosal et al., 2021a). Therefore, we define the context of the current utterance as the preceding utterances $\mathbf{u}_j, \forall j < i$. Previous research has focused on modeling intra- and inter-speaker context dependencies to capture emotional dynamics in conversations (Poria et al., 2019b). To enhance the contextual information, we employ a graph network that utilizes different edge types guided by ACK to integrate contexts.

### 3.4.1 Graph Construction

Unlike the previous graph construction (Ghosal et al., 2019; Li et al., 2021; Tu et al., 2022b), they incorporate future utterances to update the current utterance node. Following Shen et al. (2021b), we build a directed graph $\mathcal{G} = (\mathcal{V}, \mathcal{E}, \mathcal{R}, \mathcal{A})$ to model

context dependencies. $\mathbf{u}_i \in \mathcal{V}$ and $\mathbf{r}_k \in \mathcal{R}$ represent the utterance node and edge type, respectively. $\mathbf{e}_{i,j} = (\mathbf{u}_i, \mathbf{r}_k, \mathbf{u}_j) \in \mathcal{E}$ denotes the edge between node $i$ and $j$, where $j > i$. The weight of $\mathbf{e}_{i,j}$ is denoted as $\boldsymbol{\alpha}_{i,j} \in \mathcal{A}$. The utterance-level encoder initializes each node $\mathbf{u}_i$. The edge types in $\mathcal{R}$ are determined by the speaker and various ACK.

### 3.4.2 Information Aggregation

For each node at layer $\xi$, the graph context encoder aggregates its neighboring nodes as follows.

$$\widehat{\mathbf{H}}_i^{(\xi)} = \sum_{j \in \mathbf{N}_r} \mathbf{M}_s(\mathbf{Q}_j^{(\xi)}) + \mathbf{M}_r(\mathbf{Q}_j^{(\xi)}) \quad (4)$$

$$\mathbf{Q}_j^{(\xi)} = \boldsymbol{\alpha}_{i,j}^{(\xi)} \mathbf{W}_s^{(\xi)} \mathbf{H}_j^{(\xi)} \quad (5)$$

$$\boldsymbol{\alpha}_{i,j}^{(\xi)} = \mathbf{Softmax}_{(}\mathbf{W}_e^{(\xi)} [\mathbf{H}_i^{(\xi-1)} \oplus \mathbf{H}_j^{(\xi)}]) \quad (6)$$

$$\mathbf{H}_j^{(\xi)} = \overleftrightarrow{\mathbf{GRU}}_u(\mathbf{H}_j^{(\xi-1)}) \quad (7)$$

where $\mathbf{N}_r$ denotes the set of neighboring nodes. $\mathbf{Q}_j^{(\xi)}$ is the context representation of $u_i$. $\mathbf{W}_e^{(\xi)} \in \mathbb{R}^{d_h \times 2d_h}$ and $\mathbf{W}_s^{(\xi)} \in \mathbb{R}^{d_h \times d_h}$ represent projection parameters of the model. $\mathbf{M}_s(\cdot) = \mathbb{1}_{[\eta_i = \eta_j]}$ and $\mathbf{M}_r(\cdot) = \mathbb{1}_{[\eta_i \neq \eta_j]}$ denote the indicator function, used to model the different context dependencies. In previous methods, $\eta$ was generally equal to $\mathbf{s}$, used to model inter- and inter-speaker context representation $\widehat{\mathbf{H}}_i^{(\xi)}$. However, by utilizing ACK, $\eta$ can represent the CR relationship, CT semantic correlation, and EC2 to obtain the knowledge-enriched contextual representation $\widehat{\mathbf{KH}}_i^{(\xi)}$.

$$\mathbf{H}_i^{(\xi)} = \overleftrightarrow{\mathbf{GRU}}_h(\widehat{\mathbf{CH}}_i^{(\xi)}, \mathbf{H}_i^{(\xi-1)}) \quad (8)$$

$$\widehat{\mathbf{CH}}_i^{(\xi)} = \mathbf{W}_c^{(\xi)}[\widehat{\mathbf{H}}_i^{(\xi)} \oplus \widehat{\mathbf{KH}}_i^{(\xi)}] \quad (9)$$

$$\widehat{\mathbf{KH}}_i^{(\xi)} = \mathbf{CT}_i^{(\xi)} \oplus \mathbf{CR}_i^{(\xi)} \oplus \mathbf{EC2}_i^{(\xi)} \quad (10)$$

where $\mathbf{CT}_i^{(\xi)}$, $\mathbf{CR}_i^{(\xi)}$, and $\mathbf{EC2}_i^{(\xi)}$ denote the context representation by using ACK-enriched $\eta$. $\widehat{\mathbf{CH}}_i^{(\xi)}$ denotes the knowledge-enriched contextual representation. $\mathbf{W}_c^{(\xi)} \in \mathbb{R}^{d_h \times 4d_h}$ represents projection parameters of the model.

## 3.5 Contrastive Learning Module

For the classifier, we employ a feed-forward neural network to predict the distributions of emotions.

$$\widehat{\mathbf{Y}}_i = \mathbf{Argmax}(\mathbf{Softmax}(\mathbf{W}_h\widehat{\mathbf{O}}_i + b_h)) \quad (11)$$

$$\widehat{\mathbf{O}}_i = \mathbf{ReLU}(\mathbf{W}_r\mathbf{O}_i + b_r) \quad (12)$$

$$\mathbf{O}_i = \mathbf{H}_i^{(0)} \oplus \mathbf{H}_i^{(\xi)} \oplus \cdots \oplus \mathbf{H}_i^{(\zeta)} \oplus \mathbf{x}_i \quad (13)$$

where $\mathbf{W}_r \in \mathbb{R}^{d_h \times ((\zeta+1)d_h + 4d_u)}$ and $\mathbf{W}_r \in \mathbb{R}^{d_h \times d_h}$ are projection parameters. $b_r$ and $b_h$ are biases. $\widehat{\mathbf{Y}} \in \mathbb{R}^N$ is the predicting emotional label set, where $N$ is the total number of utterances.

$$\mathcal{L}_{ce} = \mathbf{CrossEntropy}(\widehat{\mathbf{Y}}, \mathbf{Y}) + \beta \|\Theta\|_2 \quad (14)$$

where $\mathcal{L}_{ce}$ is the classification loss. $\Theta$ is a set of projection parameters. $\beta$ represents the coefficient of $L_2$-regularization. To distinguish between ordinary utterances and SC utterances, as well as MP utterances, and to obtain vector representations for utterances related to the TP. we introduce the SCL loss items $\mathcal{L}_s$, $\mathcal{L}_m$ and $\mathcal{L}_t$, as follows.

$$\mathcal{L} = \mathcal{L}_{ce} + \psi_s\mathcal{L}_s + \psi_t\mathcal{L}_t + \psi_m\mathcal{L}_m \quad (15)$$

$$\mathcal{L}_{s/t/m} = -\frac{1}{n_b} \sum \log\left(\mathbf{P}(\mathbb{S}/\mathbb{T}/\mathbb{M})\right) \quad (16)$$

$$\mathbf{P}(\triangle) = \frac{\sum_{j=1}^{n_b} \mathbb{1}_{[i \neq j]} \mathbb{1}_{[\triangle_i = \triangle_j]} \boldsymbol{\ell}(\mathbf{O}_i, \mathbf{O}_j)}{\sum_{k=1}^{n_b} \mathbb{1}_{[i \neq j]} \boldsymbol{\ell}(\mathbf{O}_i, \mathbf{O}_k)} \quad (17)$$

where $n_b$ is the size of a mini-batch sample. $\psi_s$, $\psi_t$ and $\psi_m$ are tuned hyper-parameters. $\{\mathbb{s}_i/\mathbb{m}_i/\mathbb{t}_i\}_{i=1}^{n_b} \in \mathbb{S}/\mathbb{M}/\mathbb{T}$ indicates whether the $i$-th utterance is a satirical expression, metaphorical expression, and its topic, respectively. $\mathbb{S}/\mathbb{T}/\mathbb{M}$ is the subset of ALK, generated by ChatGPT to promote the utterance representations. $\boldsymbol{\ell}(\star, \star) = e^{simi(\star,\star)/\tau}$, where $\tau$ is the temperature parameter and $simi(.)$ denotes the cosine similarity function.

## 4 Experiments

### 4.1 Datasets

We conduct experiments on three datasets: IEMO-CAP (Busso et al., 2008), EmoryNLP (Zahiri and Choi, 2018), and MELD (Poria et al., 2019a). The dataset statistics are presented in Table 1.

| Dataset | Dialogues | | | Utterances | | |
|---|---|---|---|---|---|---|
| | train | val | test | train | val | test |
| IEMOCAP | 120 | | 31 | 5,810 | | 1,623 |
| MELD | 1039 | 114 | 280 | 9,989 | 1,109 | 2610 |
| EmoryNLP | 659 | 89 | 79 | 7,551 | 954 | 984 |

| Dataset | Classes | Metric |
|---|---|---|
| IEMOCAP | 6 | Weighted Avg. F1 |
| MELD | 7 | Weighted Avg. F1 |
| EmoryNLP | 7 | Weighted Avg. F1 |

Table 1: Statistics of experimental datasets.

**IEMOCAP** comprises dyadic sessions where actors engage in improvisations or scripted scenarios. Each utterance in this dataset is labeled with one of the following emotions: happy, angry, neutral, sad, excited, or frustrated. As there is no validation set in this dataset, we adopt the approach from Shen et al. (2021b) by utilizing the last 20 dialogues from the training set for validation.

**MELD** is a multi-party conversation dataset collected from the TV show *Friends*, which is an extension of the EmotionLines dataset (Hsu et al., 2018). Each utterance is annotated with one emotion from the set: surprise, fear, disgust, anger, sadness, neutral, or joy, along with one sentiment from the set: neutral, negative, or positive.

**EmoryNLP** consists of multi-party sessions from the TV show *Friends* and each utterance is labeled with one emotion from the set: joyful, scared, peaceful, sad, powerful, mad, or neutral, along with one sentiment, as suggested in Ghosal et al. (2020), from the set: neutral, negative, or positive.

### 4.2 Comparison Models

We compare our proposed framework with various ERC baselines, including **RNN-based models:** CauAIN (Zhao et al., 2022), COSMIC (Ghosal et al., 2020), DialogueRNN (Majumder et al., 2019); **Memory networks:** ICON (Hazarika et al., 2018), AGHMN (Jiao et al., 2020); **Graph-based models:** DialogueGCN (Ghosal et al., 2019), DAG-ERC (Shen et al., 2021b), SKAIG (Li et al., 2021); **Transformer-based models:** KET (Zhong et al., 2019), BERT_BASE (Kenton and Toutanova, 2019), RoBERTa (Liu et al., 2019); **Generative models:** CoG-BART (Li et al., 2022a), Curie (Olmo et al., 2021), ChatGPT (Ouyang et al., 2022)[2].

---

[2]Appendix A.1 details the prompt of ChatGPT and Curie for ERC, and Appendix A.2 explains the prompt used to obtain multiple knowledge mentioned in this paper.

| | Methods | IEMOCAP | MELD | EmoryNLP |
|---|---|---|---|---|
| | Baseline | 67.45 | 64.76 | 38.46 |
| **ALK** | **w/** TP | 67.74 | 64.89 | 38.72 |
| | **w/** SC | 67.64 | 65.07 | 38.56 |
| | **w/** MP | 68.26 | 65.12 | 38.63 |
| | **w/** TP + SC | ▲ 67.90 | △ 65.02 | ▲ 38.89 |
| | **w/** TP + | ∅ 67.73 | ▲ 65.16 | ▲ 38.88 |
| | **w/** SC + MP | ▲ 68.31 | ▲ 65.19 | ▲ 38.90 |
| | **w/** TP + SC + MP | △ 67.66 | ▲ 65.17 | ▲ 38.95 |
| **AUK** | **w/** EC | 67.80 | 65.13 | 38.70 |
| | **w/** CS | 67.86 | 65.16 | 39.07 |
| | **w/** ACS | 68.88 | 65.28 | 38.55 |
| | **w/** EC + CS | △ 67.85 | ▲ 65.25 | △ 38.87 |
| | **w/** EC + ACS | ∅ 67.53 | △ 65.16 | ▲ 38.72 |
| | **w/** CS + ACS | △ 67.97 | △ 65.27 | △ 39.06 |
| | **w/** EC + CS + ACS | △ 67.88 | ▲ 65.34 | △ 38.89 |
| **ACK** | **w/** CR | 68.02 | 65.15 | 38.78 |
| | **w/** CT | 67.66 | 65.19 | 38.66 |
| | **w/** EC2 | 67.68 | 65.07 | 38.79 |
| | **w/** CR + CT | △ 67.87 | ▲ 65.25 | ▲ 39.20 |
| | **w/** CR + EC2 | ▲ 68.49 | ▲ 65.28 | ▲ 38.83 |
| | **w/** CT + EC2 | ▲ 68.46 | ▲ 65.24 | ▲ 38.89 |
| | **w/** CR + CT + EC2 | ∅ 67.59 | ▲ 65.34 | ▲ 38.86 |

Table 2: Average weighted F1 score (%) of Baseline (MKFM w/o any knowledge) based on homologous knowledge. The best test scores are highlighted with an underline. ▲, △, and ∅ represent performance greater than any individual knowledge, not less than certain knowledge, and lower than any knowledge, respectively.

## 4.3 Experimental Settings

We perform a hyper-parameter search for MKFM on each dataset with a validation set, including learning rate, batch size, dropout rate, tuned hyper-parameters $\Psi_{s/t/m}$, $n_w$, and $n_l$. And we let $d_u = 1024$, $\tau = 0.07$, $d_x = 768$, and $d_h = 300$ on each dataset[3]. The reported results are all based on the average weighted F1 score of 5 runs on the test set.

## 5 Experimental Results

### 5.1 Contributions of Multiple Knowledge

Table 2 presents the performance of the model using each piece of knowledge. We can observe that almost all of the knowledge sources have a positive impact on the ERC task. Additionally, based on the results from the three datasets, we can see that for ALK, MP shows the most significant improvement on the IEMOCAP dataset, with an increase of 0.81%. For AUK, ACS exhibits the most powerful performance improvement on the IEMOCAP dataset, with an increase of 1.43%. As for ACK,

<hr>

[3]Appendix A.3 details more hyper-parameter settings.

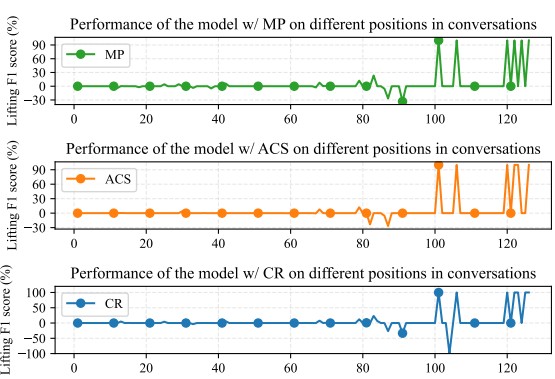

Figure 3: Lifting performance of the model on different positions in conversations.

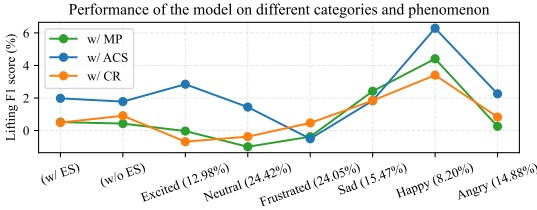

Figure 4: Lifting performance of the model on ES problems and different emotional categories.

CR demonstrates the most significant performance improvement on the IEMOCAP dataset, with a rise of 0.57%. Most importantly, to further investigate the contributions of MP, ACS, and ACS in enhancing performance, we present their effects across different emotional categories, various positions in conversations, and their impact on utterances w/ emotion shift (ES) and w/o ES.

**Contribution of knowledge to utterances in different positions:** Fig. 3 show that improvements become noticeable after the 100th utterance and become significant after the 120th utterance. However, there is a slight dip in performance between the 80th and 100th utterances, followed by a gradual recovery and stabilization. The utterances of performance enhancement are mainly observed toward the end of conversations, possibly because ERC models prioritize context in immediate proximity (Ghosal et al., 2021b) due to their similar semantics, making it challenging to detect utterances requiring longer contexts. By incorporating knowledge, the model can consider semantic elements of utterances from greater distances, which is crucial for handling long conversation data.

**Contribution of knowledge to utterances in different emotional categories:** Regarding different categories, as depicted in Fig. 4, all this knowledge

consistently boosts the performance of minority classes, such as 'happy'. On the other hand, for the majority classes, such as 'neutral', except for the model w/ ACS, leveraging other knowledge exhibits a decline in performance. Notably, the model w/ ACS achieves promising performance across all classes, except for a slight decrease in the emotion 'frustrated', which further explains its significant performance improvement. Overall, the incorporation of external knowledge appears to alleviate the issue of class imbalance in ERC.

**Contribution of knowledge to utterances in ES problems:** Concerning ES problems, although using this knowledge contributes to performance improvements, we find that the enhancements are more prominent for utterances w/o ES. This suggests that the ES problem in ERC remains a challenge, and solely relying on the introduced knowledge is not sufficient for effective resolution.

## 5.2 Complementarity of Knowledge

**Complementarity of homologous knowledge:** In Table 2, we can observe that there is a strong synergy between the homologous knowledge sources ALK and ACK. However, the complementarity performance of the model with AUK is notably poor, which could be attributed to the introduction of excessive noise associated with this knowledge source (Jiang et al., 2022; Tu et al., 2022a).

**Complementarity of non-homologous knowledge:** Moving on to non-homologous knowledge sources, in Table 3, the combination of TP performs poorly in the IEMOCAP dataset. In contrast to other knowledge types, some utterances may lack specific topic assignments during annotation. This inconsistency in annotation hinders the complementarity of the model, and this phenomenon is further amplified in long conversations. This observation holds true even when combined with three knowledge sources ALK + AUK + ACK. Furthermore, we also explore a simpler approach to integrating knowledge without considering the complementarity of heterogeneous knowledge, as shown in Table 5. This approach selects the best-performing knowledge or combination of knowledge from each homogeneous source. Experimental results show that this method further improved the MELD and EmoryNLP datasets. However, performance is slightly decreased on the long dialogue dataset IEMOCAP. This decrease can be mainly attributed to the dominance of ACS knowledge on

| | Methods | IEMOCAP | MELD | EmoryNLP |
|---|---|---|---|---|
| | Baseline | 67.45 | 64.76 | 38.46 |
| **ALK + AUK** | w/ TP + EC | ▲ 67.82 | △ 65.09 | ▲ 39.30 |
| | w/ TP + CS | ∅ 67.20 | △ 65.10 | ▲ 39.50 |
| | w/ TP + ACS | △ 67.84 | △ 65.16 | ▲ 39.22 |
| | w/ SC + EC | △ 67.70 | ▲ 65.24 | ▲ 39.33 |
| | w/ SC + CS | ▲ 68.04 | ▲ 65.25 | ▲ 39.17 |
| | w/ SC + ACS | △ 68.47 | △ 65.22 | ▲ 39.29 |
| | w/ MP + EC | △ 68.19 | ▲ 65.34 | ▲ 38.90 |
| | w/ MP + CS | ▲ 68.58 | ▲ 65.38 | ▲ 39.21 |
| | w/ MP + ACS | ∅ 68.16 | △ 65.19 | ▲ 39.27 |
| **ALK + ACK** | w/ EC + CR | ▲ 68.29 | ▲ 65.31 | ▲ 39.23 |
| | w/ EC + CT | ▲ 68.38 | ▲ 65.34 | ▲ 39.11 |
| | w/ EC + EC2 | ▲ 68.05 | ▲ 65.32 | ▲ 39.06 |
| | w/ CS + CR | ▲ 68.18 | ▲ 65.33 | ▲ 39.53 |
| | w/ CS + CT | ▲ 68.22 | ▲ 65.39 | ▲ 39.19 |
| | w/ CS + EC2 | ▲ 68.45 | ▲ 65.39 | ▲ 39.28 |
| | w/ ACS + CR | △ 68.02 | ▲ 65.38 | ▲ 39.08 |
| | w/ ACS + CT | △ 68.51 | ▲ 65.34 | ▲ 39.26 |
| | w/ ACS + EC2 | △ 68.08 | ▲ 65.50 | ▲ 39.76 |
| **AUK + ACK** | w/ TP + CR | ∅ 67.39 | ▲ 65.43 | ▲ 38.97 |
| | w/ TP + CT | ∅ 67.36 | ▲ 65.35 | ▲ 39.22 |
| | w/ TP + EC2 | ▲ 67.84 | ▲ 65.41 | ▲ 39.02 |
| | w/ SC + CR | ▲ 68.55 | ▲ 65.44 | ▲ 39.40 |
| | w/ SC + CT | ▲ 68.79 | ▲ 65.36 | ▲ 39.30 |
| | w/ SC + EC2 | ▲ 68.34 | ▲ 65.34 | ▲ 39.25 |
| | w/ MP + CR | ▲ 68.57 | ▲ 65.48 | ▲ 39.07 |
| | w/ MP + CT | ▲ 68.51 | ▲ 65.46 | ▲ 39.17 |
| | w/ MP + EC2 | ▲ 68.65 | ▲ 65.40 | ▲ 39.24 |
| **ALK + AUK + ACK** | w/ TP + EC + CR | ∅ 67.66 | ▲ 65.19 | ▲ 39.20 |
| | w/ TP + EC + CT | ∅ 67.05 | ▲ 65.26 | ▲ 39.06 |
| | w/ TP + EC + EC2 | ∅ 66.99 | ▲ 65.29 | ▲ 39.17 |
| | w/ TP + CS + CR | ∅ 67.44 | ▲ 65.29 | △ 39.04 |
| | w/ TP + CS + CT | ∅ 66.74 | ▲ 65.34 | △ 39.00 |
| | w/ TP + CS + EC2 | ∅ 66.64 | ▲ 65.27 | ▲ 39.11 |
| | w/ TP + ACS + CR | ∅ 67.34 | ▲ 65.43 | ▲ 39.07 |
| | w/ TP + ACS + CT | ∅ 67.38 | △ 65.26 | ▲ 38.94 |
| | w/ TP + ACS + EC2 | △ 67.75 | ▲ 65.37 | ▲ 38.99 |
| | w/ SC + EC + CR | ▲ 68.56 | ▲ 65.38 | ▲ 38.97 |
| | w/ SC + EC + CT | ▲ 68.26 | ▲ 65.32 | ▲ 39.12 |
| | w/ SC + EC + EC2 | ▲ 68.43 | ▲ 65.37 | ▲ 39.17 |
| | w/ SC + CS + CR | ▲ 68.42 | ▲ 65.41 | △ 39.00 |
| | w/ SC + CS + CT | ▲ 68.38 | ▲ 65.45 | ▲ 39.25 |
| | w/ SC + CS + EC2 | ▲ 68.32 | ▲ 65.47 | △ 39.05 |
| | w/ SC + ACS + CR | △ 68.26 | ▲ 65.51 | ▲ 39.56 |
| | w/ SC + ACS + CT | △ 68.65 | ▲ 65.46 | ▲ 39.44 |
| | w/ SC + ACS + EC2 | △ 68.22 | ▲ 65.47 | ▲ 39.34 |
| | w/ MP + EC + CR | ▲ 68.27 | ▲ 65.41 | ▲ 38.97 |
| | w/ MP + EC + CT | ▲ 68.33 | ▲ 65.37 | ▲ 39.07 |
| | w/ MP + EC + EC2 | △ 68.17 | ▲ 65.40 | ▲ 38.96 |
| | w/ MP + CS + CR | ▲ 68.53 | ▲ 65.36 | ▲ 39.25 |
| | w/ MP + CS + CT | ▲ 68.69 | ▲ 65.38 | ▲ 39.45 |
| | w/ MP + CS + EC2 | △ 68.00 | ▲ 65.44 | ▲ 39.45 |
| | w/ MP + ACS + CR | △ 68.41 | ▲ 65.46 | ▲ 39.52 |
| | w/ MP + ACS + CT | △ 68.43 | ▲ 65.44 | ▲ 39.28 |
| | w/ MP + ACS + EC2 | △ 68.67 | ▲ 65.47 | ▲ 39.61 |

Table 3: Average weighted F1 score (%) of Baseline based on non-homologous knowledge.

| | Element 1 (→) | Element 2 (←) | #Ratio |
|---|---|---|---|
| **ALK** | SC + MP → SC 94.69% | SC + MP ← SC 95.20% | 0.9946 |
| | SC + MP → MP 94.69% | SC + MP ← MP 95.20% | 0.9946 |
| **AUK** | CS + ACS → CS 91.54% | CS + ACS ← CS 91.54% | 1.0000 |
| | CS + ACS → ACS 91.54% | CS + ACS ← ACS 91.89% | 0.9962 |
| **ACK** | CR + EC2 → CR 94.13% | CR + EC2 ← CR 94.82% | 0.9928 |
| | CR + EC2 → EC2 94.40% | CR + EC2 ← EC2 95.70% | 0.9865 |
| **ALK + AUK** | MP + CS → MP 91.87% | MP + CS ← MP 93.03% | 0.9875 |
| | MP + CS → CS 91.96% | MP + CS ← CS 94.66% | 0.9714 |
| **ALK + ACK** | SC + CT → SC 95.13% | SC + CT ← SC 95.48% | 0.9964 |
| | SC + CT → CT 94.23% | SC + CT ← CT 95.26% | 0.9892 |
| **AUK + ACK** | ASC + EC2 → ASC 94.13% | ASC + EC2 ← ASC 92.72% | 1.0152 |
| | ASC + EC2 → EC2 92.61% | ASC + EC2 ← EC2 92.44% | 1.0019 |
| **ALK + AUK + ACK** | MP + CS + CT → MP 94.67% | MP + CS + CT ← MP 94.84% | 0.9982 |
| | MP + CS + CT → CS 94.49% | MP + CS + CT ← CS 96.23% | 0.9819 |
| | MP + CS + CT → CT 93.59% | MP + CS + CT ← CT 94.44% | 0.9910 |

Table 4: Complementarity of the model based on homologous and non-homologous knowledge. → and ← correspond to elements 1 and 2, respectively. A smaller ratio of elements 1 and 2 indicates stronger complementarity of the model using this knowledge.

| Methods | IEMOCAP | MELD | EmoryNLP |
|---|---|---|---|
| Baseline | 67.45 | 64.76 | 38.46 |
| **w/** ALK + AUK | 68.27 | 65.41 | 39.43 |
| **w/** ALK + ACK | 68.66 | 65.66 | 38.84 |
| **w/** AUK + ACK | 68.67 | 65.34 | 39.61 |
| **w/** AUK + AUK + ACK | 68.01 | 65.54 | 38.84 |

Table 5: Performance of the model based on the combination of best homologous knowledge.

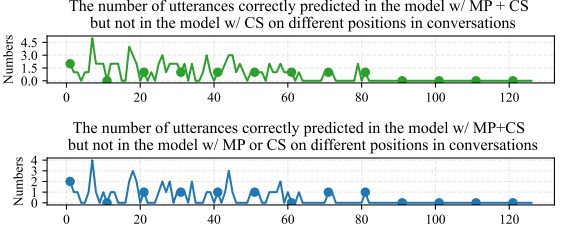

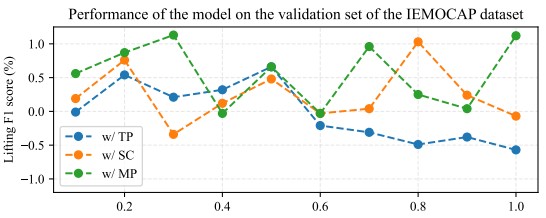

Figure 5: Lifting performance of the model on different positions in conversations.

Figure 6: Lifting performance of the model based on different coefficients of loss items.

using single or multiple knowledge (TT) by those correctly predicted using multiple knowledge alone (TF). Element 2 represents the consistency of the model by dividing the number of TT by those correctly predicted using single knowledge alone (FT). Ideally, higher consistency and lower diversity are desired, indicating smaller ratios of Element 1 to 2 representing better complementarity. The reported results in Table 4 highlight the most significant performance improvement from different types of knowledge and their combinations. The results indicate that using MP + SC yields a stronger complementary effect, while ASC + EC2 shows the least favorable complementarity.

**Further analysis to the complementarity of knowledge:** To provide a specific analysis, a set of utterances is obtained that could be correctly recognized by the model with SC but not by the model with MP + SC. Among these utterances, 27 have no ES, while 31 have ES. The similar numbers indicate that ES has a negligible impact on model consistency. However, a significant portion of these utterances (70%) belong to the 'frustrated' and 'happy' classes, indicating challenges in ensuring effective consistency for extreme classes during the fusion process. Additionally, in Fig. 5, we visualize the utterances correctly identified by MP + SC but not by SC alone, as well as the samples correctly identified by MP + SC but not by SC or MP alone. Interestingly, the distribution of these samples did not show significant changes,

that dataset, which makes simple concatenation ineffective in complementing other knowledge. This aspect is further demonstrated in the complementary heterogeneous knowledge.

**Metrics for complementarity of knowledge:** To quantify the complementarity of knowledge sources, a ratio of two elements was introduced. Element 1 denotes the diversity of the model by dividing the number of correctly predicted samples

suggesting that the diversity of the model with MP + SC primarily arises from coordinating the two types of knowledge rather than supplementing a distinct knowledge type. This further confirms the strong complementarity between the two knowledge sources.

## 5.3 Analysis of Loss Coefficients

In Fig. 6, we show the impact of different loss coefficients on model performance. Increasing $\psi_s$ and $\psi_m$ results in a fluctuating pattern but overall consistent improvement in the model's performance. However, the behavior of $\psi_t$ is different. Initially, increasing its value improves the model's performance, but then it rapidly declines. Even with some improvement afterward, the performance remains lower than that of the original model. This is because not every utterance is assigned a specific topic. Increasing $\psi_t$ amplifies the semantic gap between utterances with and without a topic, leading to a degradation in performance.

## 5.4 Comparison with Various Baselines

In Table 6, fine-tuning at the utterance level alone is inadequate for ERC due to its reliance on context and the speaker's state information. EmoBERTa, a modified version of RoBERTa, addresses this limitation by adjusting input structures and improving performance. Graph-based models outperform RNN-based models in IEMOCAP and EmoryNLP datasets by effectively capturing local context in lengthy conversations. However, in the MELD dataset (TV show data), the coherence between consecutive utterances may be lacking, reducing the advantage of graph-based models. Large models struggle in ERC, possibly due to difficulties in capturing intricate interactions, especially in lengthy conversations, as seen in their performance on the IEMOCAP dataset. And prompt-tuning the Curie model alone is insufficient for comprehensive emotional comprehension. Generative methods are not ineffective for ERC, but task-specific factors like imbalanced samples and long-term context modeling need to be considered (Li et al., 2022a).

Although LLMs often struggle to outperform smaller models in complex tasks like ERC, they can still contribute to advancements in ERC by leveraging rich knowledge. Therefore, we utilize ChatGPT to acquire diverse knowledge and proposed MKFM, a graph-based model for integrating this knowledge. We select several representative results from Table 2, 3, and 5 to conduct a compari-

| Methods | IEMOCAP | MELD | EmoryNLP |
|---|---|---|---|
| ChatGPT[§] | 40.07 | 54.37 | 37.55 |
| Curie [§] | 57.33 | 65.01 | 37.40 |
| BERT_BASE[♯] | 61.19 | 56.21 | 33.15 |
| RoBERTa[♮] | 54.55 | 62.02 | 37.29 |
| EmoBERTa[♭] | 68.57 | 65.61 | - |
| DialogueRNN[♮] | 61.21 | 56.27 | 31.70 |
| AGHMN[♭] | 62.70 | 58.10 | - |
| KET[♯] | 59.56 | 58.18 | 34.39 |
| DAG-ERC[♭] | 68.03 | 63.65 | 39.02 |
| COSMIC[♮] | 65.28 | 65.21 | 38.11 |
| SKAIG[♭] | 66.96 | 65.18 | 38.88 |
| CoG-BART[♭] | 66.18 | 64.81 | 39.04 |
| CauAIN[♭] | 67.61 | 65.46 | - |
| Baseline[§] | 67.45 | 64.76 | 38.46 |
| Baseline[§] w/ ACS | **68.88** | 65.28 | 38.55 |
| Baseline[§] w/ ALK + ACK | 68.66 | **65.66** | 38.84 |
| Baseline[§] w/ ACS + EC2 | 68.08 | 65.50 | **39.76** |

Table 6: Average weighted F1 score (%) of different methods. [§] represents our implemented model. Results with [♯] and [♮] are respectively retrieved from Zhong et al. (2019) and Ghosal et al. (2020). Results with [♭] are retrieved from the original papers.

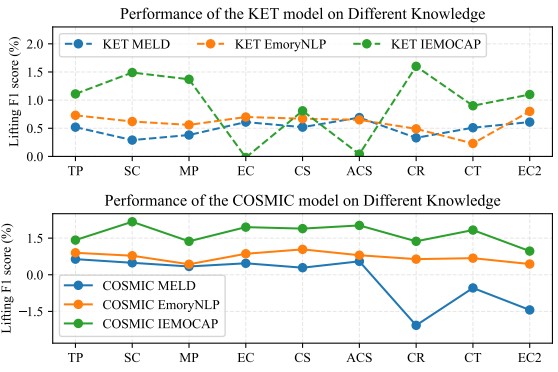

Figure 7: Lifting performance of different ERC models based on various knowledge.

son with existing methods. The results, reported in Table 6, clearly indicate that our method surpasses state-of-the-art benchmarks, which demonstrates both the feasibility of the proposed MKFM and the efficacy of such knowledge for the ERC task. Additionally, we conduct a paired t-test on the results, revealing a significant p-value ($< 0.05$) between the original baseline results and the baseline results enhanced with different knowledge.

## 5.5 Generalization Analysis

To validate the generalization of multiple knowledge in other ERC methods, we conduct experiments using the RNN-based approach COSMIC and the Transformer-based approach KET. In Fig. 7, our findings indicate that while the KET model and

ACK do not fit well in the MELD dataset, leveraging this knowledge demonstrates significant performance improvements in other cases. This observation proves the generalization ability of this knowledge for the ERC methods.

## 6  Conclusion

In this paper, we propose MKFM for integrating multiple knowledge sources in ERC, which surpasses state-of-the-art baselines. We also conduct further analysis of the contribution and complementarity of this knowledge. The **contribution of knowledge** focuses on end-of-conversation utterances, particularly those from minority classes. However, there are still unresolved ES issues. When it comes to the **complementarity of homologous knowledge**, apart from AUK's performance decline caused by noise, other cases generally show satisfactory results. As for the **complementarity of non-homologous knowledge**, the combination including TP struggles with inconsistent annotations leads to persistently poor performance in long conversations. other cases show satisfactory results. While the complementary improvement does not address ES problems effectively, it has a notable impact on utterances from minority classes, aligning with the contribution of this knowledge. Additionally, we showcase the impressive generalization ability of this knowledge in other ERC models.

## Acknowledgements

We thank the anonymous reviewers for their valuable suggestions to improve the overall quality of this manuscript. This work was partially supported by the National Natural Science Foundation of China (62006062, 62176076), Natural Science Foundation of GuangDong 2023A1515012922, Key Technologies Research and Development Program of Shenzhen JSGG20210802154400001, Shenzhen Foundational Research Funding JCYJ20200109113441941 and JCYJ20220818102415032, Guangdong Provincial Key Laboratory of Novel Security Intelligence Technologies 2022B1212010005.

## Limitations

Our approach requires diverse knowledge encompassing co-reference, topics, and emotional causes, which can be time-consuming, labor-intensive, and resource-demanding to acquire. As a result, the availability of knowledge for the MKFM model may be limited, leading to potential performance limitations. While the emergence of ChatGPT has made accessing such knowledge easier, adjusting prompts remains an unavoidable challenge, introducing a different form of "annotation" cost. Furthermore, except for AUK and ALK, the introduction of ACK necessitates certain modifications to the structure of non-graph-based models, which adds to the workload of incorporating knowledge.

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

Figure 8: Prompt template for ChatGPT.

Figure 9: Prompt template for the Curie model.

# A  Prompt Templates for ERC

In this section, we primarily focus on explaining the methodology used to obtain the results for the ChatGPT and prompt-tune Curie model in this paper. For ChatGPT, we employ a prompt template, as depicted in Fig. 8, to extract the emotion of each utterance in a conversation. This approach ensures the utilization of contextual information and facilitates the generation of well-formatted output results. On the other hand, for the Curie model, we directly access OpenAI's API for implementation. We utilize the prompt example presented in

Figure 10: Prompt template for TP.

Figure 11: Prompt template for SC.

Figure 12: Prompt template for MP.

Figure 13: Prompt template for EC.

Fig. 9 to perform prompt-tuning, which enhances the model's performance in generating emotionally appropriate responses.

# B  Prompt Templates for Knowledge

In this section, we explain the methodology used to obtain multiple knowledge. We mainly use various prompt templates (Figs. 10-18) to extract the relevant knowledge from each utterance in a conversation. Due to the diversity of outputs from ChatGPT, we also introduce some cases to standardize the format of generated knowledge such as CR, as shown in Fig. 16. Especially, we conduct a manual evaluation of extracted knowledge. Specifically, we randomly select 10% conversations to evaluate the quality of this knowledge. Three human evaluators rate each extracted knowledge on a scale of 0 to 5: 0 for extremely poor, 1 for poor, 2 for fair, 3 for good, 4 for very good, and 5 for excellent quality. The weighted average of these scores determines the knowledge quality. Only knowledge with an average score above 3 is retained. Otherwise, adjust the prompt template if necessary, re-extract, and reevaluate.

You are an invaluable assistant in analyzing commonsense knowledge in each utterance in the conversation. Commonsense Knowledge refers to people's normal, general Knowledge of the everyday world.Commonsense knowledge is the general knowledge about things, behaviors, relations and events that people accumulate in their daily life.It is the basis of understanding and anticipation of the environment in which people live.For example, it is commonly known that fire is hot, water is wet, and a sad person may cry.

The formats are as follows:
Input format :
sentence index. Speaker : sentence

Your reply format:
{'#1':'commonsense knowledge','#2':'commensense knowledge','#3':'commensense knowledge'...}
For each utterance, reply a sentence of commonsense knowledge limited 25 words
You must answer in the format I gave you.
You should reply nothing but the format I gave you.

Figure 14: Prompt template for CS.

You are an invaluable assistant in analyzing affective commonsense knowledge in each utterance in the conversation.

Affective commonsense knowledge refers to the common cognitive understanding of emotions and their expression. It involves people's general knowledge of understanding, expressing, and communicating emotions. Affective commonsense knowledge includes people's understanding of emotional expressions, responses and changes in social interactions. For example, people generally know that laughter usually indicates happiness and crying usually indicates sadness.

The formats are as follows:
Input format :
utterance index. Speaker : sentence

Your reply format:
{'#1':'affective commonsense knowledge','#2':'affective commonsense knowledge','#3':'affective commonsense knowledge'...}

For each utterance, reply a sentence of affective commonsense knowledge limited 25 words.

Figure 15: Prompt template for ACS.

Your work is to find an entity (pronoun or noun or noun phrase) with antecedents (pronoun or noun or noun phrase) co-referring to the entity, \
which means the entity and the antecedents refer to the same underlying real-world entities. Please answer the entities in the current utterance, the antecedents and the ids of utterances where the antecedents are located.

Please answer the entities in the current utterance, the antecedents and the ids of utterances where the antecedents are located.

Now I give you some task examples:
For example :
#1. Speaker 0 : Hello, nice to meet you.
#2. Speaker 1 : I am fine, thank you.
#3. Speaker 0 : Tom is eating fish.
#4. Speaker 1 : So am I! I also like eating it like him.

the correct answer is :
{
  'Speaker 0 | #1' : [
     [ 'you' , '#2' ] ,
  ] ,
  'Speaker 1 | #2' : [
     [ 'you' , '#1' ] ,
     [ 'I' , '#2' ] ,
     [ 'I' , '#4' ]
  ] ,
  'Tom | #3' : [
     [ 'Tom' , '#3' ] ,
     [ 'him' , '#4' ]
  ] ,
  'fish | #3' : [
     [ 'fish' , '#3' ] ,
     [ 'it' , '#4' ]
  ]
}

Figure 16: Prompt template for CR.

In a conversation, we can understand the above each utterance according to its relevant history utterances. Write your answer in the form of id of the each utterance: {ids of most relevant historical utterances}.

Figure 17: Prompt template for CT.

In a conversation, the emotions of the current utterance can be influenced by specific utterances before it. Your job entails identifying an index of utterances that may be the emotional cause of the current utterance. Write your answer in the form of : id of the each utterance: {ids of utterances that affect the emotions of the current utterance}.

Figure 18: Prompt template for EC2.

## C  Hyper-parameter settings

We perform a hyper-parameter search for Baseline (MKFM w/o any knowledge) on each dataset with a validation set, including learning rate, batch size, dropout rate, and so on. The seeds are 200, 201, 202, 203, and 204. The detailed search results of hyper-parameters are shown in Table 7. Additionally, each training and testing process is run on a

| Methods | IEMOCAP | MELD | EmoryNLP |
|---|---|---|---|
| Learning rate | 0.0005 | 0.00001 | 0.0005 |
| Batch size | 16 | 8 | 32 |
| Dropout rate | 0.2 | 0.1 | 0.3 |
| The number of layers $\zeta$ | 6 | 2 | 2 |
| $\psi_t$ | 0.5 | 0.3 | 0.2 |
| $\psi_s$ | 0.8 | 0.5 | 0.5 |
| $\psi_m$ | 1.0 | 0.8 | 0.8 |

Table 7: Search results of hyper-parameters.

single NVIDIA A100-PCIE-40GB GPU and the reported results are all based on the average score of 5 random runs on the test set.

## D  Glossary of acronyms

In this section, we compile a thorough glossary of acronyms, as shown in Table 8, to assist readers in comprehending the concepts in the paper.

| Acronym | Explanation |
| --- | --- |
| ERC | Emotion Recognition in Conversations, the research task of this paper. |
| ES | Emotion Shift, where two consecutive utterances in a conversation exhibit different emotions. |
| SCL | Supervised Contrastive Learning, a training methodology for classification tasks. |
| MKFM | Multiple Knowledge Fusion Model, integrates three different knowledge: Utterance-level Encoder for AUK, Graph Context Encoder for ACK, and Contrastive Learning module for ALK. |
| AUK | Auxiliary Utterance Knowledge, a category of knowledge represented by one or several sentences for each utterance in a conversation. |
| CS | CommonSense knowledge of utterances, a type of AUK. |
| ACS | Affective CommonSense knowledge of utterances, a type of AUK with a focus on emotions and sentiments. |
| EC | Emotional Cause of utterances, a type of AUK focusing on context clues or triggers giving rise to emotions and sentiments. |
| ACK | Auxiliary Context Knowledge, a category of knowledge represented by an index list for each utterance in a conversation. |
| CR | Co-Reference relationships between utterances, a type of ACK represented as an index list of historically related utterances. |
| EC2 | Emotional Cause relationships between utterances, a type of ACK represented as an index list indicating emotional causes behind the current utterance. |
| CT | Context for a better understanding of utterances, a type of ACK represented as an index list of historically relevant utterances. |
| ALK | Auxiliary Label Knowledge, a category of knowledge represented by a label for each utterance in a conversation. |
| TP | Topics of utterances, a type of ALK represented as the topic label of each utterance in a conversation. |
| SC | Sarcasm indication for utterances, a type of ALK denoted by labels 1 or 0. |
| MP | Metaphor indication for utterances, a type of ALK denoted by labels 1 or 0. |

Table 8: Glossary of acronyms.