# OpenReview forum: "An Empirical Study on Multiple Knowledge from ChatGPT for Emotion Recognition in Conversations"
_EMNLP/2023/Conference — EMNLP 2023 Findings_

### Official Review · Reviewer_9VVj · 2023-08-03

**Soundness:** 3

**Excitement:**

3: Ambivalent: It has merits (e.g., it reports state-of-the-art results, the idea is nice), but there are key weaknesses (e.g., it describes incremental work), and it can significantly benefit from another round of revision. However, I won't object to accepting it if my co-reviewers champion it.

**Paper Topic And Main Contributions:**

The purpose of this paper is to acquire valuable insights (such as co-reference, topics, emotional causes, etc.) to improve the task of emotion recognition in conversation. In order to accomplish this, the authors introduce a Multiple Knowledge Fusion Model (MKFM) that integrates diverse knowledge sources to enhance the performance of emotion recognition. Subsequently, they perform a series of experiments and analyses using established emotion datasets.

**Reasons To Accept:**

- The idea presented in this paper is intriguing and warrants the investigation.
- The extensive number of experiments and analyses illustrates the advantages of leveraging diverse knowledge sources for the task of emotion recognition.

**Reasons To Reject:**

- One of the primary drawbacks of this paper is the excessive use of abbreviations, which creates confusion for readers and requires constant backtracking.

- Another issue is the mention of utilizing ChatGPT for obtaining multiple knowledge sources, without a dedicated section describing its methodology. Given that ChatGPT is referenced in multiple sections of the paper (e.g., title and introduction), it would have been beneficial to provide a thorough explanation of its role in the methodology.

- Although the paper discusses the various categories of knowledge sources in the third paragraph of the introduction, there is insufficient description of these sources in the methodology section and how the proposed model (MKFM) effectively incorporates them. To address this, I recommend following the structure outlined in the introduction, dividing the methodology section into three categories: 1. Auxiliary Contextual Knowledge, 2. Auxiliary Label Knowledge, and 3. Utterance Knowledge. This approach would help readers understand how the proposed model integrates different types of knowledge and facilitate the connection between the three auxiliary knowledge types.

In short, this paper is limited by inadequate clarification of the use of ChatGPT, excessive use of abbreviations, and lack of organizational structure.

-------------------------------------------------------------
I thank the authors for providing a detailed response. While I appreciate the inclusion of a glossary of abbreviations, I would recommend reducing the number of abbreviations to only a few. This will enhance the paper's readability and ease of navigation. I am pleased to note that the authors have addressed the remaining comments I had.

**Reproducibility:**

3: Could reproduce the results with some difficulty. The settings of parameters are underspecified or subjectively determined; the training/evaluation data are not widely available.

**Reviewer Confidence:**

4: Quite sure. I tried to check the important points carefully. It's unlikely, though conceivable, that I missed something that should affect my ratings.

---

> ### Author Rebuttal · Authors · 2023-08-28
>
> We would like to express our great appreciation for your valuable comments to improve the quality of this manuscript.
>
> **Answer of Reject#1:**
> &ensp;&ensp;Thanks for your valuable comment. We will include a comprehensive glossary of acronyms in the appendix, as shown below, to assist readers in comprehending the concepts in the paper. Thank you again for your insightful input; we believe this addition will significantly enhance the clarity and comprehensibility of our paper.
> | Acronym | Explanation                                                                                                  |
> |:---------:|:--------------------------------------------------------------------------------------------------------------:|
> | ERC     | ERC represents Emotion Recognition in Conversations, which is also the research task of this paper.                              |
> | ES      | ES represents Emotion Shift, where two consecutive utterances in a conversation exhibit different emotions.                   |
> | SCL     | SCL represents Supervised Contrastive Learning, which is a training methodology for classification tasks.                          |
> | MKFM    | MKFM represents Multiple Knowledge Fusion Model, which is used to integrate three different knowledge. Specifically, MKFM employs the Utterance-level Encoder to integrate AUK, the Graph Context Encoder to incorporate ACK, and utilizes the Contrastive Learning module to fuse ALK. |
> | AUK     | AUK represents Auxiliary Utterance Knowledge, which refers to a category of knowledge represented by one or several sentences, corresponding to each utterance in a conversation.                    |
> | CS      | CS belongs to AUK and represents CommonSense knowledge of utterances. Unlike other forms of knowledge like triplets, obtaining utterance knowledge is simpler.                                |
> | ACS     | ACS belongs to AUK and represents Affective CommonSense knowledge of utterances. ACS is similar in form to CS but with a greater emphasis on knowledge related to emotions and sentiments.                     |
> | EC      | EC belongs to AUK and represents Emotional Cause of utterances. EC is also similar in form to CS but focuses more on context clues or triggers that give rise to emotions and sentiments.                                  |
> | ACK     | ACK represents Auxiliary Context Knowledge, which refers to a category of knowledge represented by an index list, corresponding to each utterance in a conversation.                    |
> | CR      | CR belongs to ACK and represents Co-Reference relationships between utterances. CR is an index list of historical utterances with a co-reference relationship to the current utterance.              |
> | EC2     | EC2 belongs to ACK and represents Emotional Cause relationships between utterances. EC2 is an index list of historical utterances indicating the emotional causes behind the current utterance.               |
> | CT      | CT belongs to ACK and represents Context for a better understanding of utterances. CT is an index list of historical utterances that help in understanding the current utterance.          |
> | ALK     | ALK represents Auxiliary Label Knowledge, which refers to a category of knowledge represented by a label, corresponding to each utterance in a conversation.                            |
> | TP      | TP belongs to ALK and represents Topics of utterances. TP is the topic label of each utterance in a conversation.                                                      |
> | SC      | SC belongs to ALK and represents whether the utterance is expressing SarCasm or not, typically denoted by the labels 1 or 0.                                    |
> | MP      | MP belongs to ALK and represents whether the utterance is expressing MetaPhor or not, typically denoted by the labels 1 or 0.                                 |
>
> **Answer of Reject#2:**
> &ensp;&ensp;Thanks for your valuable comment. We have revised the manuscript to include a detailed section devoted to the methodology of utilizing ChatGPT for acquiring multiple knowledge. In this process, the most crucial aspect is to obtain a suitable prompt template for each type of knowledge. Therefore, we randomly selected 10% of conversations to evaluate the quality of this knowledge. Three human evaluators rated each extracted knowledge on a scale of 0 to 5: 0 for extremely poor, 1 for poor, 2 for fair, 3 for good, 4 for very good, and 5 for excellent quality. The weighted average of these scores determined the knowledge quality. Only knowledge with an average score above 3 was retained. Otherwise, adjust the prompt template if necessary, re-extract, and reevaluate. Next, we can utilize the prompt templates to acquire this knowledge, as demonstrated in the example given in Figure 1. We will detail it in the revision.
>
> **Answer of Reject#3:**
> &ensp;&ensp;Thanks for your insightful comment. In this paper, MKFM employs the Utterance-level Encoder to integrate AUK, the Graph Context Encoder to incorporate ACK and utilizes the Contrastive Learning module to fuse ALK. To emphasize the details of knowledge integration, we will restructure the methodology section according to the framework outlined in the introduction, as per your advice. We appreciate your attention to the clarity and structure of our methodology section. Your comment has been invaluable in guiding us toward a more comprehensive and reader-friendly presentation of our work.

---

### Official Review · Reviewer_7phw · 2023-08-04

**Soundness:** 4

**Excitement:**

4: Strong: This paper deepens the understanding of some phenomenon or lowers the barriers to an existing research direction.

**Missing References:**

--

**Paper Topic And Main Contributions:**

The authors propose a Multiple Knowledge Fusion Model (MKFM) to effectively integrate Multiple knowledge (e.g., co-reference, topics, emotional causes, etc) generated by prompting LLMs for Emotion Recognition in Conversations (ERC).

**Questions For The Authors:**

Are you sure that the context data obtained from ChatGPT is correct and not subject to hallucination?
Have you thought of manually validating at least some of them? To assess its quality and reliability?
Do you think the approach is generalizable with a good level of reliability?

(issues are effectively resolved by authors)

**Reasons To Accept:**

- Technically Robust
- Exhaustive Experimental Session
- Innovative model is proposed
- Trending and relevant topic
- New research frontiers

**Reasons To Reject:**

- Complex to read and follow
- Results are not statistically validated (issues are effectively resolved by authors)
- Only one dataset (issues are effectively resolved by authors)
- Low replicability (issues are effectively resolved by authors)

**Reproducibility:**

3: Could reproduce the results with some difficulty. The settings of parameters are underspecified or subjectively determined; the training/evaluation data are not widely available.

**Reviewer Confidence:**

4: Quite sure. I tried to check the important points carefully. It's unlikely, though conceivable, that I missed something that should affect my ratings.

**Typos Grammar Style And Presentation Improvements:**

--

---

> ### Author Rebuttal · Authors · 2023-08-28
>
> We would like to express our great appreciation for your valuable comments to improve the quality of this manuscript.
>
> **Answer of Reject#1:**
> &ensp;&ensp;Thanks for your valuable comment. We will include a comprehensive glossary of acronyms in the appendix, as shown below, to assist readers in comprehending the concepts in the paper. Thank you again for your insightful input; we believe this addition will significantly enhance the clarity and comprehensibility of our paper.
> | Acronym | Explanation                                                                                                  |
> |:---------:|:--------------------------------------------------------------------------------------------------------------:|
> | ERC     | ERC represents Emotion Recognition in Conversations, which is also the research task of this paper.                              |
> | ES      | ES represents Emotion Shift, where two consecutive utterances in a conversation exhibit different emotions.                   |
> | SCL     | SCL represents Supervised Contrastive Learning, which is a training methodology for classification tasks.                          |
> | MKFM    | MKFM represents Multiple Knowledge Fusion Model, which is used to integrate three different knowledge. Specifically, MKFM employs the Utterance-level Encoder to integrate AUK, the Graph Context Encoder to incorporate ACK, and utilizes the Contrastive Learning module to fuse ALK. |
> | AUK     | AUK represents Auxiliary Utterance Knowledge, which refers to a category of knowledge represented by one or several sentences, corresponding to each utterance in a conversation.                    |
> | CS      | CS belongs to AUK and represents CommonSense knowledge of utterances. Unlike other forms of knowledge like triplets, obtaining utterance knowledge is simpler.                                |
> | ACS     | ACS belongs to AUK and represents Affective CommonSense knowledge of utterances. ACS is similar in form to CS but with a greater emphasis on knowledge related to emotions and sentiments.                     |
> | EC      | EC belongs to AUK and represents Emotional Cause of utterances. EC is also similar in form to CS but focuses more on context clues or triggers that give rise to emotions and sentiments.                                  |
> | ACK     | ACK represents Auxiliary Context Knowledge, which refers to a category of knowledge represented by an index list, corresponding to each utterance in a conversation.                    |
> | CR      | CR belongs to ACK and represents Co-Reference relationships between utterances. CR is an index list of historical utterances with a co-reference relationship to the current utterance.              |
> | EC2     | EC2 belongs to ACK and represents Emotional Cause relationships between utterances. EC2 is an index list of historical utterances indicating the emotional causes behind the current utterance.               |
> | CT      | CT belongs to ACK and represents Context for a better understanding of utterances. CT is an index list of historical utterances that help in understanding the current utterance.          |
> | ALK     | ALK represents Auxiliary Label Knowledge, which refers to a category of knowledge represented by a label, corresponding to each utterance in a conversation.                            |
> | TP      | TP belongs to ALK and represents Topics of utterances. TP is the topic label of each utterance in a conversation.                                                      |
> | SC      | SC belongs to ALK and represents whether the utterance is expressing SarCasm or not, typically denoted by the labels 1 or 0.                                    |
> | MP      | MP belongs to ALK and represents whether the utterance is expressing MetaPhor or not, typically denoted by the labels 1 or 0.                                 |
>
> **Answer of Reject#2:**
> &ensp;&ensp;Thanks for your valuable comment. We performed a paired t-test on the experimental outcomes presented in Table 6. The results indicate a p-value of < 0.05 between the original baseline results and the baseline results enhanced with different knowledge. Even in the scenario with the smallest observed enhancement—specifically, between the baseline with ACS and the baseline on the EmoryNLP dataset—the p-value stood at 0.03, highlighting statistical significance. We will detail it in the revision.
>
> **Answer of Reject#3:**
> &ensp;&ensp;Thanks for your insightful comment. In our study, we have carried out experiments utilizing three distinct datasets: IEMOCAP, EmoryNLP, and MELD. The detailed statistics pertaining to these datasets are presented in Table 1.
>
> **Answer of Reject#4:**
> &ensp;&ensp;Thanks for your insightful comment. We will release the source code after the acceptance of this paper to facilitate reproducibility by readers.
>
> **Answer of Questions:**
> &ensp;&ensp;Thanks for your valuable questions. The concerns regarding the accuracy of the context data and the hallucination issue are indeed crucial considerations. To mitigate the risks of errors and the hallucination issue, we conducted a manual evaluation of extracted knowledge.
> &ensp;&ensp;Specifically, we randomly selected 10% of conversations to evaluate the quality of this knowledge. Three human evaluators rated each extracted knowledge on a scale of 0 to 5: 0 for extremely poor, 1 for poor, 2 for fair, 3 for good, 4 for very good, and 5 for excellent quality. The weighted average of these scores determined the knowledge quality. Only knowledge with an average score above 3 was retained. Otherwise, adjust the prompt template if necessary, re-extract, and reevaluate. We will detail it in the revision.
> &ensp;&ensp;In Section 5.5, we validated the generalizability of our approach, as depicted in Figure 7. Our results show performance improvements in different ERC models, which utilize the MKFM to integrate this knowledge derived from ChatGPT. This observation confirms the capacity of our approach to generalize across various ERC methods.

---

### Official Review · Reviewer_poob · 2023-08-04

**Soundness:** 3

**Excitement:**

3: Ambivalent: It has merits (e.g., it reports state-of-the-art results, the idea is nice), but there are key weaknesses (e.g., it describes incremental work), and it can significantly benefit from another round of revision. However, I won't object to accepting it if my co-reviewers champion it.

**Paper Topic And Main Contributions:**

The paper tackles the problem of emotion recognition in conversations. The authors introduce "multiple knowledge" such as co-reference, topics of the utterance, the emotional cause, sarcasm, metaphor, etc. as features to improve the detection of emotion in conversation. They acquire the extra knowledge information through ChatGPT prompts which they engineer in a way to extract the information in their particular format. They categorize the knowledge information based on different literature research.

The introduced model is a pretrained Roberta which encodes the utterances and enhances the utterance representations with the auxiliary information they obtained from the ChatGPT output. The model also employs graph networks to capture contextual information in the conversation. Their model graph utilizes information such as coreference and emotional cause to represent relationships between nodes. The classifier used for predicting the emotion in conversation employs contrastive learning.

The authors run several experiments with different settings for the multiple knowledge they annotate to conversation emotion detection datasets. They conclude that generally speaking the model benefits from the incorporated extra knowledge. They show, however, that the improvements fluctuates as the number of utterances increases but the performance is typically better towards the end of the conversation.

They also illustrate the effect of the extra knowledge on the prediction of emotional categories and show that the contribution of knowledge boosts the performance of the minority classes in the datasets such as 'happy'.
They compare their model's performance with other baselines and they show that their model surpasses the state-of-the-art benchmarks on the datasets they use for experiments.

**Questions For The Authors:**

A. How exactly is the contrastive learning part of the model used? Is it used to extract information such as sarcasm and metaphor to incorporate as part of the extra knowledge used to predict the emotion of the utterance?

B. Why this particular typology of knowledge information? Can the authors introduce their own knowledge categorization and make the prompt engineering easier?

C. How do you validate the output of the ChatGPT?

**Reasons To Accept:**

The paper has the following strengths:

1. The authors overcome the problem of annotating the conversation datasets by employing ChatGPT prompts which they engineer to extract the needed knowledge information. This approach is very promising for other NLP tasks to compensate for the lack of gold-standard annotations specially for challenging tasks.

2. The paper includes examples of the prompts used for producing the multiple knowledge categories used in training the emotion detection model. The prompt templates can be useful for future research.

3. The in-depth analysis of how each category of the extra knowledge has helped in improving the model's performance as well as which emotional categories has been positively affected.

4. The authors show that their model's performance surpasses a number of benchmarks.

**Reasons To Reject:**

There a few issues which the authors may want to consider:

1. The long list of acronyms used in the paper makes it very difficult to read. It would be very helpful if there is a glossary of acronyms in the appendix if the authors insist on using all these acronyms to explain their experiments.

2. The difference between the three knowledge categories is not very clear. Some of the categories seem to relate to each other and are not really distinguishable, others seem to be difficult to understand. The authors relied on the categorization of other research from different domains but some of the labeling and their meanings are not clear.

3. The contrastive learning module section is difficult to understand.

4. Although the approach of obtaining knowledge from ChatGPT prompting is a good idea, there is no mention of any type of validation for the produced information. The extra knowledge can constitute noise rather than incorporate extra features.

5. The improved performance is marginal with different settings. The paper needs to justify the extra effort that is exerted to obtain this knowledge as compared to the marginal improvement in detecting emotions in the conversation datasets used.

**Reproducibility:**

3: Could reproduce the results with some difficulty. The settings of parameters are underspecified or subjectively determined; the training/evaluation data are not widely available.

**Reviewer Confidence:**

3: Pretty sure, but there's a chance I missed something. Although I have a good feel for this area in general, I did not carefully check the paper's details, e.g., the math, experimental design, or novelty.

**Typos Grammar Style And Presentation Improvements:**

Please check language on lines:

079, 106-107

---

> ### Author Rebuttal · Authors · 2023-08-28
>
> We would like to express our great appreciation for your valuable comments to improve the quality of this manuscript.
>
> **Answer of Reject#1:**
> &ensp;&ensp;Thanks for your valuable suggestion. We will include a comprehensive glossary of acronyms in the appendix, as shown below, to assist readers in comprehending the concepts in the paper. Thank you again for your insightful input; we believe this addition will significantly enhance the clarity and comprehensibility of our paper.
> | Acronym | Explanation                                                                                                  |
> |:---------:|:--------------------------------------------------------------------------------------------------------------:|
> | ERC     | ERC represents Emotion Recognition in Conversations, which is also the research task of this paper.                              |
> | ES      | ES represents Emotion Shift, where two consecutive utterances in a conversation exhibit different emotions.                   |
> | SCL     | SCL represents Supervised Contrastive Learning, which is a training methodology for classification tasks.                          |
> | MKFM    | MKFM represents Multiple Knowledge Fusion Model, which is used to integrate three different knowledge. Specifically, MKFM employs the Utterance-level Encoder to integrate AUK, the Graph Context Encoder to incorporate ACK, and utilizes the Contrastive Learning module to fuse ALK. |
> | AUK     | AUK represents Auxiliary Utterance Knowledge, which refers to a category of knowledge represented by one or several sentences, corresponding to each utterance in a conversation.                    |
> | CS      | CS belongs to AUK and represents CommonSense knowledge of utterances. Unlike other forms of knowledge like triplets, obtaining utterance knowledge is simpler.                                |
> | ACS     | ACS belongs to AUK and represents Affective CommonSense knowledge of utterances. ACS is similar in form to CS but with a greater emphasis on knowledge related to emotions and sentiments.                     |
> | EC      | EC belongs to AUK and represents Emotional Cause of utterances. EC is also similar in form to CS but focuses more on context clues or triggers that give rise to emotions and sentiments.                                  |
> | ACK     | ACK represents Auxiliary Context Knowledge, which refers to a category of knowledge represented by an index list, corresponding to each utterance in a conversation.                    |
> | CR      | CR belongs to ACK and represents Co-Reference relationships between utterances. CR is an index list of historical utterances with a co-reference relationship to the current utterance.              |
> | EC2     | EC2 belongs to ACK and represents Emotional Cause relationships between utterances. EC2 is an index list of historical utterances indicating the emotional causes behind the current utterance.               |
> | CT      | CT belongs to ACK and represents Context for a better understanding of utterances. CT is an index list of historical utterances that help in understanding the current utterance.          |
> | ALK     | ALK represents Auxiliary Label Knowledge, which refers to a category of knowledge represented by a label, corresponding to each utterance in a conversation.                            |
> | TP      | TP belongs to ALK and represents Topics of utterances. TP is the topic label of each utterance in a conversation.                                                      |
> | SC      | SC belongs to ALK and represents whether the utterance is expressing SarCasm or not, typically denoted by the labels 1 or 0.                                    |
> | MP      | MP belongs to ALK and represents whether the utterance is expressing MetaPhor or not, typically denoted by the labels 1 or 0.                                 |
>
> **Answer of Reject#2:**
> &ensp;&ensp;Thanks for your insightful comment. We divide this knowledge, demonstrated effective for emotion detection, into three categories (AUK, ACK, and ALK) based on data formats. Specifically, AUK refers to a category of knowledge represented by one or several sentences corresponding to each utterance. ACK refers to a category of knowledge represented by an index list corresponding to each utterance. ALK refers to a category of knowledge represented by a label corresponding to each utterance. For instance, for SC in ALK, we can utilize ChatGPT to determine whether an utterance carries sarcasm undertones, which serves as a form of label information. CR in ACK focuses on enhancing contextual understanding. Determining if an utterance refers back to something mentioned earlier might not be applicable to SC. Furthermore, the approach of obtaining labels or list-based information for CS in AUK is unconventional, making it more suitable for aiding the semantic understanding of utterances.
> &ensp;&ensp;Intuitively, AUK, ACK, and ALK have distinct roles in ERC. AUK enhances the semantics of utterances, ACK enriches contextual information, and ALK introduces labels for various tasks to learn task-specific latent features, thus improving the model's emotional understanding capability. Additionally, more details on the labeling and their meanings can be found in the **Answer of Reject#1**. The provided example in Figure 1 enhances the clarity of our illustration for AUK, ACK, and ALK.
>
> **Answer of Reject#3:**
> &ensp;&ensp;Thanks for your insightful comment. For integrating ALK, the obvious approach is to employ multi-task learning. However, this could lead to increased workload and additional training complexity, which may overly complicate the baseline. Hence, we leverage Supervised Contrastive Learning (SCL), which pulls together utterance representations with the same label and pushes apart those with different labels. This is achieved without escalating model complexity and raising training expenses. It only adds an additional loss term while still delivering effective results.
>
> **Answer of Reject#4:**
> &ensp;&ensp;Thanks for your valuable comment. In the process of extracting knowledge from ChatGPT, noise does indeed emerge. The noise introduced by irrelevant knowledge has consistently remained one of the challenges in ERC. Even without using ChatGPT to acquire knowledge, similar noise is inevitable (Tu et al., 2022a).
> &ensp;&ensp;To mitigate the impact of noise, we conducted a manual evaluation of extracted knowledge. Specifically, we randomly selected 10% conversations to evaluate the quality of this knowledge. Three human evaluators rated each extracted knowledge on a scale of 0 to 5: 0 for extremely poor, 1 for poor, 2 for fair, 3 for good, 4 for very good, and 5 for excellent quality. The weighted average of these scores determined the knowledge quality. Only knowledge with an average score above 3 was retained. Otherwise, adjust the prompt template if necessary, re-extract, and reevaluate. We will detail it in the revision.
>
> *Tu, G., Liang, B., Jiang, D., & Xu, R. (2022). Sentiment-Emotion-and Context-guided Knowledge Selection Framework for Emotion Recognition in Conversations. IEEE Transactions on Affective Computing.*
>
> **Answer of Reject#5:**
> &ensp;&ensp;Thanks for your insightful comment. As for why it is worth expending the extra effort to acquire this knowledge, we can elaborate based on the following key points.
> 1. Some settings as shown in Table 6 show significant improvements, highlighting the potential for outperforming strong baselines. Further optimizing the method of knowledge integration could lead to additional performance improvement.
> 2. Although the improved performance is marginal with different settings on the whole, it becomes significant after the 100th utterance as shown in Figure 3, effectively compensating for the limitations of ERC models in modeling long-term contextual dependencies.
> 3. The incorporation of external knowledge appears to alleviate the issue of class imbalance in ERC as depicted in Figure 4.
> 4. The strong generalizability of this knowledge is evident from Figure 7. Our results demonstrate significant performance improvements in different ERC models that leverage this knowledge.
>
> **Answer of Question A:**
> &ensp;&ensp;Thanks for your insightful question. Contrastive learning module, as mentioned in this paper, is primarily applied to integrate ALK rather than knowledge extraction. Knowledge extraction for SC (or MP) is achieved using ChatGPT. This knowledge belongs to ALK and represents whether the utterance is expressing sarcasm (or metaphor) or not, typically denoted by the labels 1 or 0. In the contrastive learning module, we leverage Supervised Contrastive Learning (SCL) to pull together utterance representations with the same label and to push apart those with different labels.
>
> **Answer of Question B:**
> &ensp;&ensp;Thanks for your insightful question. We categorize this knowledge based on its data format due to our consideration of knowledge attributes. Specifically, AUK refers to a category of knowledge represented by one or several sentences corresponding to each utterance. ACK refers to a category of knowledge represented by an index list corresponding to each utterance. ALK refers to a category of knowledge represented by a label corresponding to each utterance. For instance, for SC in ALK, utilizing ChatGPT to determine whether an utterance carries sarcasm undertones is intuitively reasonable, and serves as a form of label information. CR in ACK focuses on enhancing contextual understanding. Determining if an utterance refers back to something mentioned earlier might not be applicable to SC. Obtaining labels or index lists for CS in AUK is unconventional, making it more suitable for enhancing the semantics of utterances.
> &ensp;&ensp;Additionally, introducing our own knowledge categorization for facilitating prompt engineering is a promising and theoretically feasible concept. Thank you again for your insightful input; We will attempt this approach to simplify prompt engineering in the source code upon paper acceptance.
>
> **Answer of Question C:**
> &ensp;&ensp;Thanks for your valuable question. We conducted a manual evaluation of extracted knowledge. Specifically, we randomly selected 10% of conversations to evaluate the quality of this knowledge. Three human evaluators rated each extracted knowledge on a scale of 0 to 5: 0 for extremely poor, 1 for poor, 2 for fair, 3 for good, 4 for very good, and 5 for excellent quality. The weighted average of these scores determined the knowledge quality. Only knowledge with an average score above 3 was retained. We will detail it in the revision.
>
> **Answer of Typos Grammar Style And Presentation Improvements:**
> &ensp;&ensp;Thanks a lot for your valuable suggestions, we have corrected them in the revision and will avoid these mistakes in the future.

---

### Meta-Review · Area_Chair_1dxp · 2023-09-17

**Recommendation:** 3

**Metareview:**

- The paper uses many abbreviations and it is hard to read the manuscript.
- An excessive number of experiment is done but the results are marginally better
- The model is complex and that makes reproducibility a challenging task
- The method is tested only on one dataset

---

### Decision · Program_Chairs · 2023-10-07

**Decision:**

Accept-Findings

**Comment:**

- The paper uses many abbreviations and it is hard to read the manuscript.
- An excessive number of experiment is done but the results are marginally better
- The model is complex and that makes reproducibility a challenging task
- The method is tested only on one dataset